# Structural transformation and political economy: A new approach to inclusive growth

**Madiha Kamal**[1]\*, **Ambreen Fatima**[2]

**1** Management Studies Department, Bahria University, Islamabad Campus, Karachi, Pakistan, **2** Applied Economics Research Centre, University of Karachi, Karachi, Pakistan

\* madihakamal@live.com

**Data Availability Statement:** The data was obtained from the Handbook of Statistics on Pakistan Economy (https://www.sbp.org.pk/departments/stats/PakEconomy_HandBook/index.htm) and varies economic survey of Pakistan (https://www.finance.gov.pk/survey_2022.html).

## Abstract

The "Jobless growth" of Pakistan's economy and its acquaintances with frequent changes in different governments in power (both democratic and military regime) and its influence on economic growth persuades to conduct a qualitative analysis of the structural transformation on the disaggregated level to determine whether or not it has any connection or impact on different sectors of the economy. The fundamental objective of this study is to (a) explore how sectoral transformation has affected economic development and employment, as well as their relationships, and (b investigate how these dynamics have impacted the generation of sustainable employment in the economy. For this reason, the study takes sectoral transformation into account. The study applies the sectoral employment Elasticity technique for this objective. The results of our evaluation indicate that the economy functions better under military regimes than it does during democratic periods. The cause might be anything, but our goal is to determine the employment elasticity of Pakistan's main political powers. As an agriculture-based economy, most of the population relies on the agriculture sector; hence, its decline highlights the need for reforms. If this sector doesn't grow by creating additional jobs, it will lose its employment power.

## Introduction

The period beginning in 1993–1994 and ending in 2011–2012 is recognised as a "Jobless Growth" period due to the degeneration of the elasticity of GDP's employment growth, and it is characterised by high GDP growth without proportional employment growth, as determined by a review of the available literature from previous studies Bhattacharya [1]. The reason was the economic liberalization and other reforms such as relaxed labour market regulations, trade liberalization and emergence of technology-intensive services. These above mentioned were some of the leading reasons for weaker job creation (Ghose [2]; Sen [3]; Eichengreen [4]. The structural restructuring of the economy is another factor that might be regarded a primary cause of low GDP growth. As explained by Kuznets [5], it is the reallocation of economic activities from the primary sector to other sectors, and it is one of the essential elements of sustainable economic growth. Explain this process as the movement of labour inputs from low productivity to high value-adding industries and the withdrawal of surplus

**Funding:** This research did not receive any specific grant from funding agencies in the public, commercial, or not-for-profit sectors.

**Competing interests:** Authors declare that there are no competing interests.

**Abbreviations:** Agr, Agriculture; Ind, Industry; Ser, Service; Emp, Employment; Output, Output; MIN, Mining & Quarrying; MAN, Manufacturing; EGW, Electricity and Gas Distribution; CON, Construction; WRT, Wholesale and Retail Trade; TSC, Transport, Storage and Communication; FI, Finance and Insurance; RS, Real State; OS, Other Services.

labour from the conventional sector Lewis [6]. With the assistance of this shift in labour inputs, the agriculture sector is converted from subsistence farming to a capitalist mode of production, which boosts agricultural labour productivity and reduces excess labour. This surplus labour is absorbed by a burgeoning modern manufacturing sector that focuses on maximising returns to scale through greater productivity and compensation Kaldor [7].

Importantly, some surplus labour may quit the labour market during the early stages of transformation. One argument is that the young spend more time learning the skills required in today's economy, prompting them to stay in school longer. Furthermore, some women may withdraw from the labour market during the early stages of the transition due to an increased need for domestic care or a lack of employment opportunities, but subsequently re-enter the labour market Goldin [8].

## Objective of the study

The basic objective of this study is to examine the sustainable employment and economic growth in Pakistan. For that purpose, this study has considered the share in growth and employment of different sector of the economy. Furthermore, the impact of different political regimes on all three sectors i.e agriculture, industrial and services sector has also been evaluated. This study is arranged in the following manner. Section 1 presents the detailed introduction of the topic. Section 2 consists of the trends of economic evaluations with facts and figures presented to show the linkage of growth and employment. Section 3highlights the existing literature review. section 4 contains the methodological framework and the data used for this study. Section 5 comprises the results of the study, and section 5 contains the conclusion of the study.

## Background of the study

Countries with high population expansion, such as Pakistan, must establish policies that can not only promote economic growth, but also accommodate many labour forces simultaneously. Rapidly expanding countries aspire to achieve sustainable employment for long-term economic growth. In developing nations with a growing youth population, such as Pakistan, it is the government's primary responsibility and greatest challenge to generate new work possibilities for new entrants and the existing labour force based on their talents and abilities.

One of the biggest challenges for Labour abundant countries like Pakistan is to follow the policies which can accommodate the huge number of people providing them with the right skill set to compete in the market. There is a strong relationship between employment and output growth it was highlighted by Okun law (1970). The study estimated that with one per cent increase in employment the GNP can be increased by 3 percentage. This fact is surely irrefutable, that with more employment opportunities, the overall output growth of the country leads to economic stability. The two most important factors to measure the economic growth of any economy from worker's perspectives are the growth of employment and growth of wages it was said by Pleic [9]. To identify a better growing economy, we must look for its employment creation capability and how much growth in wages has been increased over the years. With constant increasing Labour force, Pakistan is facing an enormous underutilization of its labour force, but it is more important to create jobs than to raise wages said by Pleic [9]. Considering the importance of employment creation, it is imperative to work on the ideas which can help new influx in the market to get absorbed. No economy can cope up from the high rise of the population without putting efforts on job creation. Highlighted the importance of human capital by focusing on the issue that for Pakistan the ideas which can accommodate the human capital must be kept under consideration for the long-run economic growth this study

conducted by Zulfiqar [10]. Polices that are capable enough to absorb swiftly growing population with the right choice of job should be given the importance as they are the engine for the economy. Panda [11] study showing declining trend in the Indian employment growth rate in the agriculture sector. Negative employment growth in manufacturing sector. But the output growth is significant in industry and service sectors and agriculture sector output growth is low.

Likewise, the structural transformation of labour force form low productive sector to high productive sector with systematic way is undeniable. For long-term economic growth, it plays a very crucial role. For economic growth and development, it is very significant for the authorities to open the untapped ventures for the new arrivals in the labour market. These new entries in the labour market either skilled or unskilled are always highly motivated to give their best in the beginning of their career. They put extra effort and always ready to go to extra mile to prove their abilities. What is needed is the right direction and guidance for them to take the best of their abilities.

## Structural transformation in Pakistan

In case of Pakistan, it is being witnessed that after 1960's Green Revolution, when the country's agriculture sector transformed from an average to a highly productive sector, it was the agriculture sector that dominated the country's productivity. With a large share of GDP, it was believed during that time that landlords are the most successful individuals. After that time, the greater use of mechanisation and the availability of inexpensive raw materials for the garment industry led to the development of new industries. With abundant availability to cotton, Pakistan's clothing sector flourishes by expanding production and contributing positively to the GDP. In the meantime, as the industrial sector expanded, it contributed by default to the transformation of unskilled labour into human capital. With the fundamental technical knowledge required to function in any business, individuals were being trained to operate machines.

Meanwhile, trained human capital adapted to the services industry. Differently. Human capital in the services sector accounts for a substantial share of GDP. Although the agriculture sector is still the leader in terms of engaging more people, its proportion in GDP has declined significantly over the years, suggesting that merely engaging more people in any area is not enough to create output in the economy, but making them productive is. It includes agricultural disguised employment. As the services sector contributes a little share of GDP, but its labour absorptive capacity is low, and Pakistan is a labor-abundant country, the most important thing for any country with a large population is to have as many people working as possible. Unfortunately, Pakistan is not meeting this criteria. Low-income Pakistanis were particularly affected by structural changes, which cut their salaries. Unskilled suffer most.

Exogenous technology growth benefits all three industries. With the rapid rise in technology, the services sector now relies heavily on information technology and related sectors. Demand and supply drove inter-sectoral change. Unemployed workers go from farm to manufacturing. While there, they learned some basic abilities to go from unskilled to skilled labour, and their capacity to adapt helped them.

Rural-urban migration also drives sectoral change. Small-town residents move to larger cities for better pay. This forces people to leave farming or service work. Domestic help, drivers, tailors, Maison, plumbers, and others are examples. They moved to the city to earn, gain new skills, and work. We cannot ignore the earnings from the above-mentioned Gray economy services, even though they are not included in GDP. Wage differentials and labour demand drive sector shifts. For higher wages, workers transfer from low-wage to high-wage markets.

Does structural transformation matter? According to Arthur Lewis, countries that transition from agro-based to advanced economic activity move from poverty to rapid growth, but

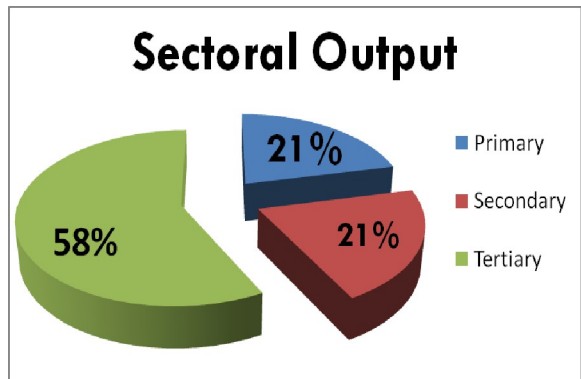
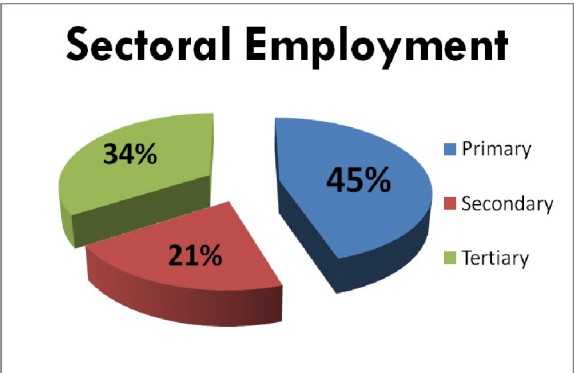
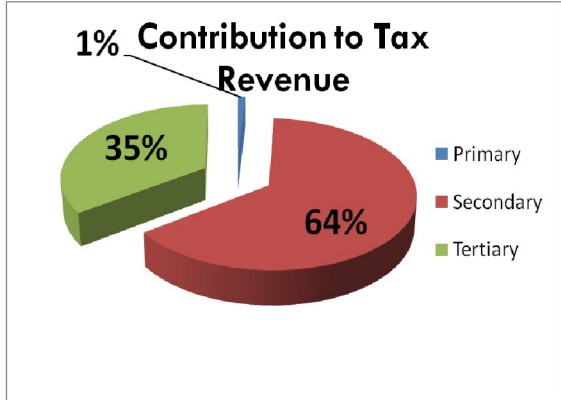

**Fig 1. Snapshot of Pakistan economy and its sectoral structure.**

speed is key. In Pakistan, it matters. Increased economic activity raises income, which helps impoverished people buy basics and improve their level of living. Humans rationally think better when full. With more money, people can consider other wants. These small income gains enhance GDP and economic growth. Pakistan has slower structural changes. Manufacturing stagnates while services improve after the 1970s. Pakistan's main industry was agriculture, and workers in it didn't need any training. They usually engage family members to help them, creating disguised employment. Results demonstrate that agricultural employment is high, but income and output are low (Fig 1). The services industry is slowly increasing output and productivity. Statistics reveal a large and improving services sector. An hour is needed to increase structural transformation with systematic alterations.

M R Bilal [12] found in "The shift to services sector: Prospects of dynamics in Pakistan" that while the services sector input is expanding, its capacity to engage more human capital is decreasing. Since Pakistan is an agro-based country and the agriculture and manufacturing sectors are interconnected, the author stated that government action might strengthen the role of the manufacturing sector to reduce over-employment in agriculture and underemployment in services. Due to its scope, the study does not explain structural transformation.

Modernization is hampered by a lack of infrastructure, financial means, and modern sources. All of the above-mentioned amenities make it easier for an economy to accelerate its growth and find new growth opportunities.

Pakistan must employ the newest structural transformation strategies for future economic growth because reallocating resources has been shown worldwide to drastically enhance

output and productivity. Pakistan's job market is suffering from population growth. Every year, hundreds of new workers enter the workforce. It's hard to accommodate so many with the finest option. First, turn this raw labour force into human capital, then fit it into the labour market. Pakistan, a labor-rich nation, develops and promotes labor-absorbing programmes. Pakistan's progress may be aided by fresh approaches and the region's untapped possibilities.

(Fisher [13]; Clark [14]; and Fourastié [15] introduced three-sector theory. Economic expansion shifts economic activity. It can also vary widely among sectors and subsectors. Both cases include sectoral structural change or transition. A new, concise research explores sectoral structural transformation. Does separating the economy into big sections suffice, or do we need a more comprehensive method that goes beyond economic growth factors? Sectoral structure change affects economic development?

Structural change drives economic development, according to the research. Allocating agricultural resources to other sectors reduces poverty, boosts productivity, and raises an economy's average income. Sectoral transformation pace drives development. The reallocation of labour and other economic resources from lower to higher is the foundation of economic growth literature.

Sectoral change has been neglected in Pakistan for years. Poor policies lead to failure. In this case, smooth and systematic structural reform should relocate unproductive or disguised workers from one sector to another. It requires government intervention. The agricultural sector has employed the most people. Agriculture has stagnated for years.

In Pakistan, sectoral reform is still in its early stages. Detaching oneself to relocate to another sector for greater pay is still a difficult undertaking for many people. It takes a lot of guts to step outside of one's comfort zone and learn a new trade or move from one industry to another. One of the most prevalent examples is Pakistan's agriculture industry, where people's incomes are barely enough to keep them alive. Still, they remain committed to it for the sake of their pride, as leaving behind a family farm to work for someone else is no longer acceptable anymore. The need is here to let people recognise that, despite pouring every effort into that piece of land to fit their wants, they are still unable to meet their needs.

We have noticed some really overpowering situations of sectoral employment in Pakistan using trends research spanning the years 1964 to 2018. These trends depict the genuine picture of employment in Pakistan by sector.

### Linking growth and employment–Some facts

Time starting from 1964 till 2018, we have witnessed some very overwhelming conditions of sectoral employment in Pakistan. These trends reveal Pakistan's sector-wise employment.

(Fig F1 in S1 Appendix) shows GDP growth and sectoral employment in Pakistan from 1964 to 2018. The study divided agriculture, industry, and services into subsectors for a deeper investigation. Bifurcation highlights industry growth in the graph. In our analysis, agriculture has consistently provided the most jobs. GDP growth is variable. GDP growth fluctuates. All sectors are growing. Manufacturing employs second-most people after agriculture. Real estate has the slowest growth. Pakistan recently gave the real estate company industry status and announced incentives to stimulate business.

(Fig F2 in S1 Appendix) shows GDP growth and employment output ratio. The agriculture industry has the largest employment output ratio, but its trends are dropping, indicating a decline in job generation. Trade and business rank second in employment output ratio. Manufacturing and public administration figures are also rising. Our data shows that labour is leaving agriculture for commerce, business, manufacturing, and construction. Pakistan is developing, thus the tendencies are predicted.

(Fig F3 in S1 Appendix) shows the three sectors' employment output ratio and GDP growth, illustrating that the agriculture sector, the largest industry and services sector, is declining. After the 1970s green movement in Pakistan, it declined.

## Agriculture sector

Employment created per unit of economic growth is measured by employment elasticity. The broad body of research indicates that pro-poor economic growth is appropriate with low elasticities in the agriculture sector and strong elasticities in the other sectors (Ramoni-Perazzi [16]. Agriculture underpins economic prosperity. It drives economic prosperity and national progress Sertoglu [17]. G [18] said agriculture will determine long-term economic prosperity, Economists and development specialists argue how this contributes to economic prosperity. Highly populated countries like Pakistan need such measures to increase labour force participation and growth. Those policies may assist solve unemployment problems. Agriculture has employed the most people. Agriculture has the highest ratio, which reduces efficiency and output. The sector has created jobs, but productivity has not. Some researchers debate how the agricultural sector affects economic growth Chebbi [19], Gardner [20], Lavorel et al. [21] examined the causation between agricultural value added per worker and GDP per capita to answer Gardner 2005's "is agriculture an engine of growth" question. The study found that agriculture value added drives emerging country growth. Developed nations were unclear. It confirms that developing nations have relied on agriculture.

(Fig F4 in S1 Appendix) compares GDP growth to the agriculture sector, which has grown rapidly during the 1980s. This is good and above 20% growth after 2005.

(Fig F5 in S1 Appendix) shows the agriculture employment output ratio and GDP growth declining. Labor has left the agriculture sector in the past year.

## Industrial sector

Manufacturing development changes a country's social and economic structure and raises people's living standards by improving production and technology Ellahi, N. [22]. Since the industrial revolution, this industry has driven global expansion. Indonesia, Korea, Taiwan, and China are developing countries. Better industrial development in the above countries reduced poverty and boosted economic growth Kniivila [23]. The industrial sector can enhance a country's balance of payments, exports, and import substitution by making significant changes in the real sector, which boosts national income and jobs Saeed [24].

(Fig F6 in S1 Appendix) The industrial sector is seeing a fall in terms of employment output ratio and GDP growth. Although the industrial sector increased initially, it then began to decline. Low or no understanding of the latest technology, inadequate production standards, and inexpensive substitutes from imported goods are some of the causes for the industrial sector's slow growth. The industrial industry has not implemented the most recent trendy procedures that can improve the quality of its production and allow it to compete with lower-cost goods on the market.

## Services sector

The services sector is vital to the economy, directly contributing to GDP and job creation and providing vital inputs that impact investment climate, growth, and development Dossani [25]. One thing that has emerged with wide unanimity from the literature is that this increasing chunk of the services sector in total output and employment is a characteristic of high-income elasticity of demand for services, reducing relative productivity of services, contracting out services from firms in the manufacturing sector, and increased international trade in services.

Nayyar [26]. Pakistan's services sector is booming. Male and female education rates may have increased. (Fig F7 in S1 Appendix) illustrates GDP growth and service sector employment output ratio.

## Sustainable development and economic growth

To achieve sustainable development and long-term economic growth, resources must be used in a way that benefits the country and its people. The most essential issue is how a country is raising its people's level of living from low-income, low-standard to high-income, modern. It's been assessed by how a country's economic, political, and social policies are improving its people. Economic growth drives development. Economic development involves job creation, production, and resource utilisation, while exhaustive economic growth involves productivity, R&D, innovation, economic reforms, and job creation. Economic development can be defined as the appointed mobilisation of human, organisational, financial, physical, and natural resources to improve competitive services and goods and increase their quantity for the community.

Furthermore, other elements affect a nation's economic progress Lankauskiene, Tvaronaviciene. Economic growth and development aim to accelerate asset generation. Because a nation's well-being depends on it, every nation strives for maximum results and development Tvaronavičienė, [27]. Sustainable development is complicated. It's broad because it could be related to a given's competitiveness, but it's also practical because sustainable development is measured by a wide range of metrics.

The 1987 Bruntland Commission report confirmed the concept of sustainable development as "development that meets the demands of the present without jeopardising the ability of future generations to meet their own needs" (United Nations General Assembly, 1987, p. 43). Economic growth occurs when all products and services are produced more. Economic growth ignores human capital. Most emerging nations prioritise GDP growth over sustainable development. Without sustainable development, economies can expand short-term, but long-term growth requires it. Sustainable economic growth requires strategic policies. Economics and sustainable development literature is informative and beneficial. Economic growth cannot support progress. Typically, sectoral growth and structural transformation drive economic growth. "Is sectoral reform enough for economic growth?" Is sectoral structural transformation too long-term growth? "Is this growth sustainable?"

The literature states that agriculture, manufacturing, and services drive any nation's economy. Agriculture is ancient. Agriculture provides raw ingredients for industry, which are used in various products. After training, industrial workers might work in services. These sectors are linking raw labour to human capital. The country's long-term economic development and growth will benefit from smoother sector transitions. The most recent population census indicates that Pakistan is currently facing the process of demographic transition. Providing new employment opportunities for the upcoming young generation is the biggest challenge for policy makers and economic managers. The experts conclude that Pakistan need a stable and a higher rate of economic growth to generate adequate and compatible job prospects that can duly defuse the population bomb. The recent thesis on sectoral economics revealed that agriculture sector is overemployed while services sector is the most under-employed sector in Pakistan (Bilal A. R, 2011). Since the sectoral structural transformation from agriculture sector to industrial sector and/or services sector may facilitate the process of demographic transition smoothly.

## Literature review

Khan used input-output data and the Global Computable General Equilibrium (CGE) Model on Nepal Social Accounting Matrix 2007–08 to explore cross-sectoral backward and forward

connections of the economy. Khan, M. A [28]. The study found that Nepal's manufacturing and services sector boosts economic growth, welfare, and household income.

Ibrahim, M. [29] evaluated the structural transformation in 32 African countries from 1985 to 2015 using sample splitting estimates to study the influence of trade and financial integration openness on Africa's economy (2020). The study suggests integrating financial influences and trade to examine cross-country structural transformation discrepancies. Non-tariff obstacles are Africa's main issue, according to the study.

Service-oriented Lee, J. W., & McKibbin, W. J. [30] use an empirical general equilibrium model to analyse the effects of rapid labour productivity increase in Asian countries. Consumption and investment dynamics are shown by this model. The study found that rapid productivity growth in Asia's services sector contributes to sustained and balanced economic growth, and that a positive expansion of the durable manufacturing sector is an initial condition for greater economic growth.

The panel vector autoregression model by Kim, S., Lee, J. W., & McKibbin, W. J. [31] considered Asia's demand rebalancing and supply-side productivity changes on long-term economic growth in Asia and worldwide. The study found that labour productivity shocks have positive and durable effects on Asian output, while productivity-neutral demand rebalancing shocks do not. A global intertemporal multi-sector general equilibrium model suggested that labour productivity shocks can increase foreign GDP over time, whereas rebalancing shocks have a negative international spillover effect. Labor productivity shocks can also rebalance.

Examine China's growth during the past 40 years. China uses cross-country panel data to raise its per capita income to the level of advanced economies by improving human capital, trade openness, institutions, and investment. J.W. [32]. Shift-share analysis with industry-level data shows that China's inclusive labour productivity growth depends on manufacturing sector productivity. The services sector declined from 1980 to 2010, yet the rearrangement of the labour force from agriculture to services increased labour productivity. Due to convergence impact and structural change issues, average potential GDP growth was cut by 3% to 4%. Later, China increased institutional and productivity, particularly in services.

Lee, J. W [33] used cross-country regression to examine the Republic of Korea's economic development and catch-up during the past half-century and the US-ROK output per worker gap. Reducing per capita income relative to its potential level quickly closed the gap, boosting growth and supporting conditional convergence theory. Strong investment, reduced fertility, trade openness, improved human resources, and rule of law helped the Republic of Korea catch up to the US, according to the report. Better democratic chances slow catch-up. The state improves institutional policy elements while reaching steady-state per-worker production. Export-oriented policies and manufacturing boost productivity, but the services sector underperforms. The People's Republic of China should learn from the Republic of Korea's experience to develop its services-based economy to avoid a slowdown due to convergence effect and balance between domestic consumption and services.

China's slowing growth is connected to its rebalancing from investment and production to consumption and service. China's structural transformation is compared to Japan and South Korea's. Murach, M., & Wagner, H [34].

"Employment and economic growth nexus in Nigeria" used the Ordinary Least Square (OLS) technique before and after time series data to find that the data was associated with non-stationarity using the Sodipe [35]. The study found a positive association between employment level and economic growth in Nigeria and a negative relationship between GDP growth and employment growth. To minimise Nigerian unemployment, labor-promoting investment techniques were also suggested.

Okan's youth employment idea in Europe. The author uses a different form of Okun's law to analyse the effects of real output growth on the young unemployment rate from 15 to 24 years old. Colantonio, E., & Nico, G. [36] conclude that Romania's economy does not fit Okan's law.

The quality of employment in distinct service sub-sectors compared to the industrial sector, including pay and job security. Long-term real GDP growth affected industry-sector employment from 1991 to 2001. The study found two employment models using monthly data from April 1991 to March 2001, one using real GDP as the primary explanatory variable and the other using real GDP, five macroeconomic performance parameters, and temporal trends. OLSQ regression calculated real GDP industry-sector employment elasticities. Sawtelle, B [37] reports five "Jobless Recovery" (negative employment) industries and a wide range of employment elasticities across industrial groups (2007). Employment and social protection by studying India's household survey data from 1993–94 to 2004–05, it demonstrates that either service sub-sectors are excellent or bad employers, and greater salaries do not compensate for worse job security or protection. G. Nayyar [26] assesses.

Liberalization in India's manufacturing sector 1993–2005. The study found that banking, telecommunication, insurance, and transport changes boost manufacturing productivity, while services sector reforms helped both local and foreign manufacturers. Arnold, J. [38].

One of the best-known regularities in growth and development economics is the progressive relation between per capita income and the service sector. Study finds The service industry grew in two waves, first in low-GDP countries and then in high-GDP countries. The first wave of traditional services and the second wave of modern services—financial, computer, technical, communication, legal, advertising, and business—are all receptive to information technology and more transferable across borders. Furthermore. Post-1990 low-income nations experienced the second wave, although not all economies changed equally. Eichengreen, B., & Gupta, P [4]. found it most deceptive in open-trade democracies and those closer to major global financial centres (2013).

As the industrial sector drives economic expansion, emphasise the need of power. The endogenous growth mode study uses Auto Regressive Distributed Lag (ARDL) approach to estimate long- and short-term growth. Labor is capital, the study concluded. Pakistan's economic growth depends on energy supply and industrial sector expansion; however, electricity shortages may hurt industrial performance. Industrial strategies should address electrical issues because the sector cannot improve without it. Ellahi [22]. Consider agriculture as the solution for long-term economic growth and empirically assess the impact of the agricultural sector on Nigeria's economic growth from 1981 to 2013. Real GDP, agricultural output, and oil rents are in long-term balance. Although agricultural output boosts economic growth, the Vector Error Correction model reveals that variables adjust slowly to their long-run equilibrium path. Sertoglu, K. et al. [17] suggested long-term agriculture plans with a better budget (2017).

## Methods

As explained earlier, the uninterrupted procedure of moving the labour from lower to higher productivity sectors are considered as structural changes. In this process, (a) the labour force moves from low productivity to higher productivity sectors to earn better. (b) in this procedure, the productivity within the sector also increases. The productivity within the sector is entitled to the adaptation of the latest technologies and better management practices. In the long run, it enhances production efficiency, resulting in the reallocation of the resources attached to it. With the criteria mentioned above, our analysis aligns with Ocampo [53];

Worrall et al. [54]. It has been seen that the changes which come with time bring new opportunities which were not available previously. Initially, at the comprehensive level, there was little confusion about what industry means in the 1950s, and it includes manufacturing, mining, construction, and utilities. Among these, only manufacturing was initially considered as the main interest subject. It has created economic activity at that time and gave more output per worker compared to the agriculture and services sector. It was also witnessed at the time, that few of the tradable services and agricultural value chains had shared a comprehensive characteristic with the manufacturing ((Baumol [55]; Bhagwati [56]. Some of them also got benefits to form technological changes and productivity growth as it was achieved by manufacturing and demonstrate the accumulation of the economies of scale ((Ebling [57]; Ghani [58]. Although these industries are not precisely manufacturing, this is the very reason for emphasising the movement from lower-to-higher productivity in and across all sectors in our study. There is no argument against the economic transformation with the labour force movement from lower-to-higher productivity sector (for instance, from agriculture to manufacturing or services sector) and can decrease poverty for sustainable growth in developing economies. Still, it has been seen that not all the struggles to encourage transformational policies are not always achieve success. Many countries try to do that by implementing such policies but unfortunately gain low-quality and jobless growth,

It is the continuous closing of the productivity gap across sectors and firms and between poor and rich countries. Why productivity is important because is output per worker its show how rich and poor your country. We typically focus on two mechanisms for closing these gaps:

Raising within sector productivity growth (arising productivity in manufacturing and agriculture. That can happen either thorough innovation enhances technology) either through shifting resources or by raising the productivity of existing activities.

The current study chooses to estimate sectoral employment. The current study aims to contribute by addressing this gap as well, due to the lack of the literature on factors affecting sectoral employment.

Moving labour and other resources from lower to higher productivity sector (structural change).

Both are extremely important for achieving sustained growth in productivity. Example of agriculture labour moving to low productivity sector to high productivity sector manufacturing sector.

The study has used the elasticities to calculate sectoral employment elasticities. The definition of elasticity is the percentage change in the number of people employed in the economy, connected with the percentage change to the output of the economy, which is calculated by gross domestic product Iyanatul [59], Slimane [60], Leshoro [41]. With this broad definition, usually, two ways are utilized for the calculation of elasticities. One of them is Arc elasticity which is presented below in Eq 1:

$$\varepsilon_{it} = \left( \frac{(E_{i1} - E_{i0})/E_{i0}}{(Y_{i1} - Y_{i0})/Y_{i0}} \right) \tag{1}$$

## Data

For a better and detailed understanding, this study has calculated the sectoral employment elasticities with the gap of 3year time. The study took the data starting from 1964–67 till 2015–18, combine this complete time is spread over the 54 years of Pakistan's employment history.

This study has collected the data of employed labour force through Labour Force Survey's (LFS)(https://www.sbp.org.pk/departments/stats/PakEconomy_HandBook/index.htm)

various issues and Pakistan Economic Survey which is published on the annual basis by Federal Bureau of Statistics, government of Pakistan (Pakistan Economic Survey is published annually by the Government of Pakistan, Ministry of Finance.). This is fairly a long-time duration to understand any country's employment trends. Before we delve into the detail of the above analysis, it is important to highlight the fact which we have discussed above as that it is imperative to know that Pakistan is an agriculture-based country, and an enormous number of its population directly or indirectly are associated with it. The same is describe here. The agriculture sector is more responsive to changes in employment as compared to other two sectors (industry and services). Over the years, in the agriculture sector, the negative change in employment is witnessed six times. This is highest as compared to industrial sector and services sector. This negative change is supporting the argument that in past 54 years of Pakistan's economy, the growth has increased but unfortunately, this growth is jobless, and it has not created the employment opportunities for the new entries in labour market. Although the situation in the industrial sector and services sector is also not good, but it is slightly stable as compared to agriculture sector.

## Result

In Table 1, (Khan [28]; Kapsos [61]; Leshoro [41] (see Table 2). The whole graphic shows how any economy can have positive GDP growth, negative employment elasticity, and negative employment growth with positive productivity growth. Conversely, in an economy with negative GDP growth, negative employment elasticity matches positive employment growth and negative productivity growth. When employment elasticity exceeds one, economies with positive and negative GDP growth behave differently. If the employment elasticity is zero to one, an economy with positive GDP growth will have positive employment and productivity growth. Any economy with increased production and employment is optimal. Productivity growth and employment growth elasticity are crucial for every economy to eliminate poverty. Employment elasticity provides the quantitative aspect of employment growth, but the qualitative part should not be overlooked. Both traits should be balanced.

Table 3 shows sectoral employment elasticities calculated with a three-year gap for a better understanding. The study examined Pakistan's employment history from 1964–67 to 2015–18. Understanding a country's employment trends takes this long. Before we go into the above analysis, it is necessary to note that Pakistan is an agriculture-based country, and a large portion of its population is directly or indirectly involved in it. Identical. Agriculture responds more to employment fluctuations than the other two industries (industry and services) Ahmed [62]. Six times, farm employment has declined. This exceeds industrial and service sectors. This negative trend supports the claim that Pakistan's economy has grown for 54 years, but it hasn't created jobs for new workers. Industrial and service sectors are worse than agriculture, although they are more stable.

**Table 1.**

| Employment Elasticity | GDP growth | |
|---|---|---|
| | Positive GDP growth | Negative GDP growth |
| e < 0 | (-) employment growth | (+) employment growth |
| | (+) productivity growth | (-) productivity growth |
| 0 ≤ e ≤ 1 | (+) employment growth | (-) employment growth |
| | (+) productivity growth | (-) productivity growth |
| e > 1 | (+) employment growth | (-) employment growth |
| | (-) productivity growth | (+) productivity growth |

**Table 2. Studies on the employment intensity of growth at sectoral level.**

| Authors | Period Countries | Nb of Sectors | Approach/Methodology | Results |
|---|---|---|---|---|
| Kapsos [39] | 1991–2003 160 countries | 3 | Time series/Multivariate regression | Elasticities are between 0.3 and 0.38. In the case of Morocco this value is 0.28 (1999–2003). The service sector is the most intensive in employment |
| Ajilore and Yinusa [40] | 1990–2008 Botswana | 9 | Time series/Error-correction model | Low elasticity in all sectors, Jobless growth |
| Leshoro [41] | 1980–2011 Botswana | 3 | Time series/Error-correction model | Positive but low elasticity |
| Paul et al. [42] | 1960–2014 Cameroon | 3 | Time series/OLS | Significant elasticity for agriculture (0.65) and services (1.1), but not significant for industry |
| El-Ehwany and El-Megharbel [43] | 1980/81-2004/05 Egypt | 6 | Time series/OLS | Positive elasticity in all sectors, manufacturing and mining are the most intensive in employment |
| Misra and Suresh [44] | 1993/94-2011/12 India | 6 | Time series/OLS | Significant elasticity for the manufacturing, mining and construction sector, the latter exceeds the unit |
| Mkhize [45] | 2000Q1-2012Q4 South Africa | 8 | Time series/Error-correction model | Long-term relationship in financial and business services, manufacturing, transportation and utilities |
| Sawtelle [37] | 1991M4-2001M3 United States | 15 | Time series/OLSQ | Elasticity between 1.23 (construction) and -0.04 (durable goods manufacturing), five sectors have negative elasticities |
| Islam, I. and Nazara [46] | 1977–1996 Indonesia (provinces) | 5 | Cross section/OLS | Elasticity between 0.6 and 0.7 at the global level. Agricultural (1.05), Industry (0.60), Trade (0.92), Services (0.98), Other (0.46) |
| Tadjoeddin and Chowdhury [47] | 1993–2006 (Two periods) Indonesia (provinces) | 9 | Panel/System GMM two steps | Positive elasticity in all sectors |
| Perugini [48] | 1970–2004 Italy | 5 | Panel/Fixed effect model | The elasticity for agriculture and industry is lower than for services |
| Sassi and Goaied [49] | 1983–2010 Tunisia | 15 | Panel/Mean Group model | The most labor intensive is services sectors and exporting manufacturing industries |
| Ezzahidi and El Alaoui [50] | 1991–2011 Morocco | 20 | Arithmetic method/Arc-approach (1999–2009) | Positive but low elasticity |
| Guisan and Exposito [51] | 1995–2012 5 European countries | 4 | Pool of 5 countries: Model in levels by GLS and mixed dynamic model by LS | Elasticity: Agriculture 0.37, Industry 0.21, Building 0.41, Services higher than 0.70 |

El-Hamadi, Y., Abdouni, A., Blouaouz, K [52]

Classical microeconomics states that when consumers buy things, their demand curve rises, raising prices and causing inflation. Pakistan's economy cycles successfully so far. After this stage, microeconomic theory dictates that production must rise since increased demand drives producers to produce more. Microeconomics promotes producers to use better production variables to boost output (land, labour, and capital). In the near run, capital is fixed, but the producer can increase labour demand for higher production. Unfortunately, Pakistan is not doing this. The producer imports goods from abroad to meet consumer demand instead of producing more. This hurts jobs. Services employment has grown, but it hasn't helped agriculture or manufacturing. How can unemployed people demand more in our economy? We have a high remittance rate from relatives living abroad who send money to their families to survive.

## Sectoral elasticity and political regime

Fig 2 shows Pakistan's democratic and military regimes from 1964 to 2018. Pakistan's economy fluctuates. The graphic shows that military governments do better than democratic regimes. We're interested in Pakistan's political regimes' employment elasticities, regardless of the rationale. Dr. Faheem Jehanger Khan (https://pide.org.pk/blog/economic-growth/) stated

**Table 3. Employment elasticity.**

| Years | Agr | | | Ind | | | Ser | | | Total | | |
|---|---|---|---|---|---|---|---|---|---|---|---|---|
| | Ch. Emp | Ch. Output | Elasticity | Ch. Emp | Ch. Output | Elasticity | Ch. Emp | Ch. Output | Elasticity | Ch. Emp | Ch. Output | Elasticity |
| 1964–67 | -0.07991 | 0.115284 | -0.6932 | 0.33361 | 0.238744 | 1.397356 | 0.157973 | 0.3206 | 0.492742 | 0.042488 | 0.220786 | 0.192438 |
| 1967–70 | 0.120334 | 0.279276 | 0.430879 | 0.000225 | 0.334988 | 0.000672 | -0.06097 | 0.18019 | -0.33837 | 0.048435 | 0.244044 | 0.198467 |
| 1970–73 | 0.113784 | 0.019644 | 5.792386 | -0.0101 | 0.146424 | -0.06896 | 0.090468 | 0.166304 | 0.54399 | 0.083944 | 0.104783 | 0.801122 |
| 1973–76 | 0.036427 | 0.065362 | 0.557323 | 0.137938 | 0.147464 | 0.935399 | 0.211362 | 0.21103 | 1.001574 | 0.095634 | 0.146107 | 0.654549 |
| 1976–79 | 0.071308 | 0.086829 | 0.821239 | 0.219087 | 0.217652 | 1.006596 | 0.154636 | 0.211182 | 0.732238 | 0.120493 | 0.170214 | 0.707894 |
| 1979–82 | 0.064305 | 0.157289 | 0.408833 | 0.018572 | 0.350912 | 0.052926 | 0.120018 | 0.214599 | 0.559267 | 0.069856 | 0.222871 | 0.313437 |
| 1982–85 | 0.027319 | 0.102227 | 0.267236 | 0.105167 | 0.211304 | 0.497707 | 0.115042 | 0.266669 | 0.431405 | 0.066878 | 0.205846 | 0.324892 |
| 1985–88 | 0.087581 | 0.123839 | 0.707216 | 0.057607 | 0.30847 | 0.186752 | 0.066081 | 0.195322 | 0.338319 | 0.075297 | 0.200092 | 0.37631 |
| 1988–91 | -0.06941 | 0.155729 | -0.44569 | 0.010269 | 0.196823 | 0.052173 | 0.134123 | 0.135946 | 0.986593 | 0.005174 | 0.155216 | 0.033335 |
| 1991–94 | 0.149755 | 0.091346 | 1.639416 | -0.03548 | 0.181924 | -0.19504 | 0.086287 | 0.166711 | 0.517587 | 0.092313 | 0.151157 | 0.61071 |
| 1994–97 | -0.05845 | 0.19208 | -0.30431 | 0.162277 | 0.050634 | 3.204878 | 0.22525 | 0.142648 | 1.579065 | 0.072259 | 0.131743 | 0.548485 |
| 1997–00 | 0.172666 | 0.130476 | 1.323354 | 0.010902 | 0.123537 | 0.088249 | -0.03714 | 0.096211 | -0.38605 | 0.064166 | 0.111249 | 0.576784 |
| 2000–03 | -0.02943 | 0.019849 | -1.48269 | 0.258294 | 0.114721 | 2.251491 | 0.220849 | 0.136336 | 1.619893 | 0.106415 | 0.10109 | 1.05268 |
| 2003- | 0.192965 | 0.159411 | 1.210486 | 0.180597 | 0.356996 | 0.505878 | 0.133315 | 0.222773 | 0.598433 | 0.168346 | 0.239252 | 0.703636 |
| 2003–09 | 0.124442 | 0.094253 | 1.320305 | 0.068215 | 0.08282 | 0.823652 | 0.038108 | 0.151515 | 0.251512 | 0.081789 | 0.120885 | 0.676584 |
| 2009–12 | 0.068699 | 0.077631 | 0.884938 | 0.157421 | 0.13489 | 1.167035 | 0.069202 | 0.139692 | 0.495391 | 0.087025 | 0.12488 | 0.696872 |
| 2012–15 | -0.01533 | 0.0748 | -0.20498 | 0.117963 | 0.107593 | 1.096381 | 0.039884 | 0.154616 | 0.257957 | 0.032422 | 0.125993 | 0.25733 |
| 2015–18 | -0.01418 | 0.063727 | -0.22256 | 0.129902 | 0.15306 | 0.8487 | 0.169838 | 0.213222 | 0.796533 | 0.082632 | 0.168321 | 0.490916 |

Source: Authors' Estimation

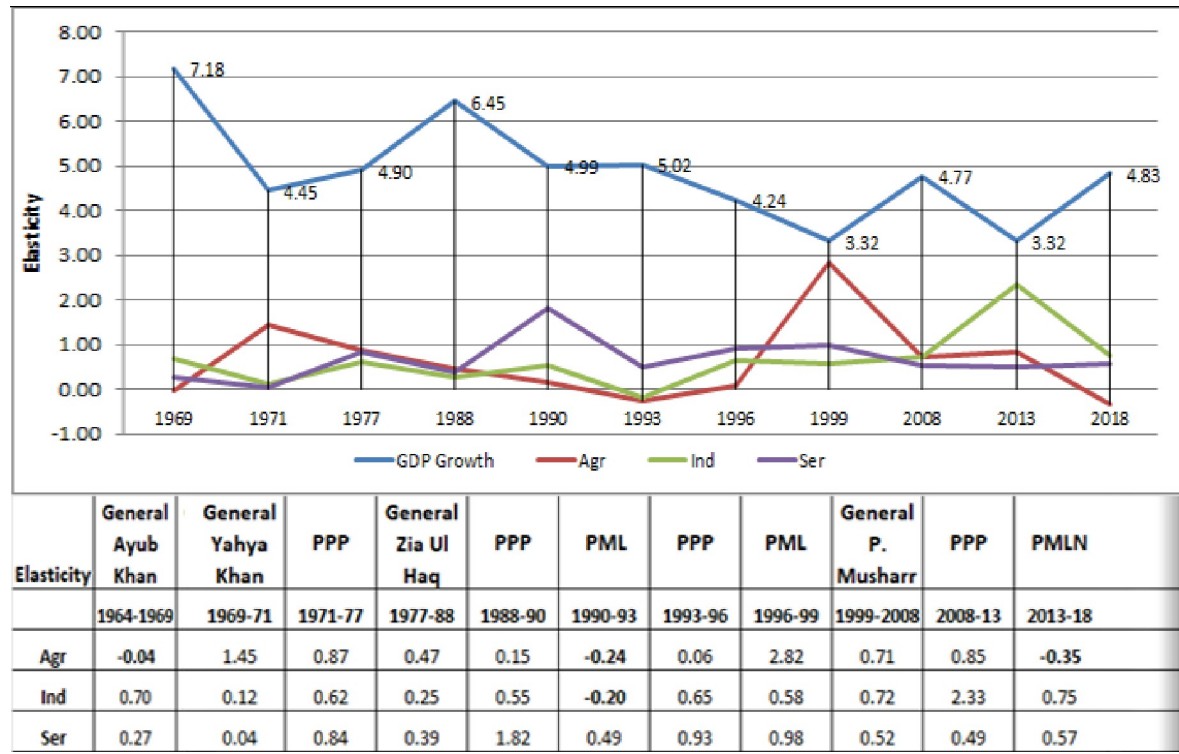

**Fig 2.** Source: Authors' Estimation.

in public policy and economic growth of Pakistan "Pakistan's economy is far behind China, India, and Bangladesh, with an average GDP growth of 4. Political uncertainty causes economic instability. Pakistan has seen four democratic eras in the past 50 years. Military involvement in governance and political party changes stunted economic growth. In these years, many action plans were created, but most were never implemented." Pakistan's sectoral elasticity and political regime are shown above for 54 years. Economic growth and political regime are strongly correlated. Pakistan's economy fluctuates. The agriculture sector's elasticity was -0.04 under General Ayub Khan's rule (1964–69). Employment in the agriculture sector is negatively affecting output. Increased output does not boost employment. Four military-democratic regimes follow. From 1969 General Yahya through the People's Party's second term, none of them showed a downward trend. No sector had negative employment in these four administrations. PMLN takes power (1990–93). The PMLN government's nine-year economic decline is dismal. PMLN ruled twice (1990–93 and 1996–99) and PPP once during the period (1993–96). Pakistan's economy has experienced the worst nine-year economic decline. The first PMLN government saw negative growth in agriculture and industry, -0.24 and -0.20, respectively. PMLN's latest stint (2013–18) saw -0.35 agricultural growth. As an agriculture-based economy, most of the population relies on the agriculture sector, hence its decline highlights the need for reforms. If this sector doesn't grow by creating additional jobs, it will lose its employment power.

Same chart. GDP rose greatest under military rule and plummeted with democracy. Whatever the reason, Pakistan's economy has only flourished under military rule.

## Conclusion and policy recommendation

Pakistan has experienced an applauding increased sectoral output growth and performance in the past military regimes. However, growth without employment especially in the agriculture and industry sector or we said good producing sectors, as well as increased levels of poverty, is a concern in Pakistan. Where the increases in sectoral output do not translate into increased sectoral employment. This study, therefore, analyzed the sectoral employment elasticity of growth in political and military regimes for Pakistan using data covering the period 1964 to 2018. The sectoral employment elasticity of growth, which is the antiquity of the correlation between employment and growth in different regimes and revealed some shocking sectoral employment circumstances in Pakistan. Followed by four military-democratic governments. None of them showed a declining tendency between General Yahya's presidency in 1969 and the People's Party's second term. In these four administrations, no sector had a negative employment rate. PMLN is elected (1990–93). The nine-year economic downturn under the PMLN government is depressing. During that time, the PMLN ruled twice (1990–1993 and 1996–1999), and the PPP once (1993–96). The economy of Pakistan has suffered the greatest nine-year drop. Agriculture and industry both saw negative growth under the first PMLN administration, with -0.24 and -0.20, respectively. Agricultural growth during PMLN's most recent term (2013–18) was -0.35. The majority of the population depends on the agriculture sector since the economy is dependent on agriculture, hence its decline emphasizes the need for changes. If this industry doesn't expand by adding jobs, it will lose its ability to employ people. This demonstrates that not only does output growth not lead to the creation of new jobs, but that employment has decreased. The cause might be related to the possibility of capital-intensive and labor-replacing processes. Furthermore, the sectoral employment elasticity of growth was quite low and it could be inferred that the growth in each of these sectors has not been labor-employment driven, but rather labor-productivity driven.

### Based on the above analysis and conclusions, several recommendations follow

Government policy should take into account employment subsidies with a stronger emphasis on young employment subsidies to boost employment placement. Other measures might include putting more labor into jobs rather than replacing it in the main industries. Additional research should look at calculating sectoral employment elasticity (or productivity) and determining how much this boosts employment in Pakistan in political regimes.

The government should give considerable consideration to creating measures to encourage the employment of the agricultural industry and its potential to integrate. Its total contribution to employment and capability for absorption employment is declining.

The industry sector have the highest elasticities in recent years. but given the industry sector's significance and the fact that it provides for the largest employment share, the government should develop policies to help it in order to lower the unemployment rate. Greater economic benefits would result from this, benefits that would go well beyond simply creating jobs.

### Limitation

Due to a lack of information, this study employed the simple Elasticity technique, which involves first building an economic framework using all social, economic, and competitiveness factors, and then using that framework to examine the estimated elasticities.

### Supporting information

**S1 Appendix.**
(DOC)

## Author Contributions

**Conceptualization:** Madiha Kamal.

**Data curation:** Madiha Kamal.

**Formal analysis:** Madiha Kamal.

**Investigation:** Madiha Kamal.

**Methodology:** Madiha Kamal.

**Supervision:** Ambreen Fatima.

**Validation:** Madiha Kamal.

**Visualization:** Madiha Kamal.

**Writing – original draft:** Madiha Kamal.

**Writing – review & editing:** Madiha Kamal.

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
