## [Decision Letter · Decision Letter 0]

26 Sep 2022

PONE-D-22-20220Structural Transformation and Political Economy: A New Approach to Inclusive GrowthPLOS ONE

Dear Dr. Kamal,

Thank you for submitting your manuscript to PLOS ONE. After careful consideration, we feel that it has merit but does not fully meet PLOS ONE’s publication criteria as it currently stands. Therefore, we invite you to submit a revised version of the manuscript that addresses the points raised during the review process.

We look forward to receiving your revised manuscript.

Kind regards,

Atif Jahanger, Ph.D

Academic Editor

PLOS ONE

Journal Requirements:

"No"

“no”

5. Please ensure that you refer to Figure 2, 3, 4, 5, 6 and 7 in your text as, if accepted, production will need this reference to link the reader to the figure.

Reviewers' comments:

Reviewer's Responses to Questions

**Comments to the Author**

1. Is the manuscript technically sound, and do the data support the conclusions?

Reviewer #1: Yes

Reviewer #2: Partly

Reviewer #3: No

2. Has the statistical analysis been performed appropriately and rigorously? 

Reviewer #1: Yes

Reviewer #2: No

Reviewer #3: No

3. Have the authors made all data underlying the findings in their manuscript fully available?

Reviewer #1: No

Reviewer #2: No

Reviewer #3: No

4. Is the manuscript presented in an intelligible fashion and written in standard English?

Reviewer #1: Yes

Reviewer #2: No

Reviewer #3: No

5. Review Comments to the Author

Reviewer #1: 1. Please rewrite abstract and elaborate more on the main findings of your analyses.

2. Describe your data and justify the time line of data selection in your write up.

3. Policy and recommendation section must include the main findings and their implication.

Reviewer #2: I read the paper very carefully and topic is very interesting. The paper have many issues for example methodology is very poor and not written properly. It is very hard to extract information from the graphs given in the text. The paper is very much lengthy and its length can be reduced by merging the graph. I think authors need to employ latest econometric techniques.

Reviewer #3: This manuscript reports on “Structural Transformation and Political Economy: A New Approach to Inclusive Growth”. I want to suggest a few suggestions to improve the manuscript's quality and better readability.

1. The English language needs more work. There are many grammatical and typo mistakes in this manuscript. The paper needs to be edited by a native English speaker.

2. The abstract should state briefly the purpose of the research, the principal results, and major conclusions. An abstract is often presented separately from the article, so it must be able to stand alone.

3. We prefer if you use the third person singular, instead of the first person singular or plural (e.g. "you, we, our").

4. The major defect of this study is the debate or argument is not clearly stated in the introduction session. Hence, the contribution is weak in this manuscript. I would suggest the author enhance your theoretical discussion and arrives at your theoretical argument.

5. The authors made this study unnecessary long and vague. Just presenting some facts and figures are not enough.

6. The copy paste diagrams are not allowed and suitable references should be provided.

7. Theoretical framework section is missing.

8. Elasticity method should be referenced.

9. See lines 687-688: the authors mentioned “(the citation must be included here)”. Please be consistent.

10. Please make sure your conclusions' section underscore the scientific value added of your paper, and/or the applicability of your findings/results, as indicated previously. Please revise your conclusion part into more details. Basically, you should enhance your contributions, limitations, underscore the scientific value added of your paper, and/or the applicability of your findings/results and future study in this session.

11. Notation" should be added to the article.

12. What are the next policy implications?

13. What are the limitations and future research agenda?

6. PLOS authors have the option to publish the peer review history of their article (what does this mean?). If published, this will include your full peer review and any attached files.

Reviewer #1: **Yes: **Faisal Abbas

Reviewer #2: No

Reviewer #3: No

---

## [Author Response · Author response to Decision Letter 0]

11 Feb 2023

Dear Reviewer,

I incorporate all your fruitful recommendations and suggestion. Thank You.

Regards.

Madiha Kamal.

---

## [Decision Letter · Decision Letter 1]

15 Mar 2023

Structural Transformation and Political Economy: A New Approach to Inclusive Growth

PONE-D-22-20220R1

Dear Dr. Kamal,

We’re pleased to inform you that your manuscript has been judged scientifically suitable for publication and will be formally accepted for publication once it meets all outstanding technical requirements.

Kind regards,

Atif Jahanger, Ph.D

Academic Editor

PLOS ONE

Additional Editor Comments (optional):

Reviewers' comments:

Reviewer's Responses to Questions

**Comments to the Author**

1. If the authors have adequately addressed your comments raised in a previous round of review and you feel that this manuscript is now acceptable for publication, you may indicate that here to bypass the “Comments to the Author” section, enter your conflict of interest statement in the “Confidential to Editor” section, and submit your "Accept" recommendation.

Reviewer #1: All comments have been addressed

Reviewer #3: All comments have been addressed

2. Is the manuscript technically sound, and do the data support the conclusions?

Reviewer #1: Yes

Reviewer #3: Yes

3. Has the statistical analysis been performed appropriately and rigorously? 

Reviewer #1: Yes

Reviewer #3: Yes

4. Have the authors made all data underlying the findings in their manuscript fully available?

Reviewer #1: No

Reviewer #3: Yes

5. Is the manuscript presented in an intelligible fashion and written in standard English?

Reviewer #1: Yes

Reviewer #3: Yes

6. Review Comments to the Author

Reviewer #1: all comments are incorporated by the author(s) that were provided by the reviewer. Therefore, I recommend accept this paper for publication in the journal PLOS One.

Reviewer #3: (No Response)

7. PLOS authors have the option to publish the peer review history of their article (what does this mean?). If published, this will include your full peer review and any attached files.

Reviewer #1: No

Reviewer #3: No

---

## [Editor Report · Acceptance letter]

3 Apr 2023

PONE-D-22-20220R1 

Structural Transformation and Political Economy: A New Approach to Inclusive Growth 

Dear Dr. Kamal:

I'm pleased to inform you that your manuscript has been deemed suitable for publication in PLOS ONE. Congratulations! Your manuscript is now with our production department. 

Kind regards, 

on behalf of

Dr. Atif Jahanger 

Academic Editor

PLOS ONE